# NERetrieve: Dataset for Next Generation Named Entity Recognition and Retrieval

Uri Katz[*1]     Matan Vetzler[*1]     Amir DN Cohen[1]     Yoav Goldberg[1,2]
[1]Bar-Ilan University     [2]Allen Institute for AI
{urikacid,matan.r.vetzler,yoav.goldberg}@gmail.com

## Abstract

Recognizing entities in texts is a central need in many information-seeking scenarios, and indeed, Named Entity Recognition (NER) is arguably one of the most successful examples of a widely adopted NLP task and corresponding NLP technology. Recent advances in large language models (LLMs) appear to provide effective solutions (also) for NER tasks that were traditionally handled with dedicated models, often matching or surpassing the abilities of the dedicated models. Should NER be considered a solved problem? We argue to the contrary: the capabilities provided by LLMs are not the end of NER research, but rather an exciting beginning. They allow taking NER to the next level, tackling increasingly more useful, and increasingly more challenging, variants. We present three variants of the NER task, together with a dataset to support them. The first is a move towards more fine-grained—and intersectional—entity types. The second is a move towards zero-shot recognition and extraction of these fine-grained types based on entity-type labels. The third, and most challenging, is the move from the recognition setup to a novel *retrieval* setup, where the query is a zero-shot entity type, and the expected result is all the sentences from a large, pre-indexed corpus that contain entities of these types, and their corresponding spans. We show that all of these are far from being solved. We provide a large, silver-annotated corpus of 4 million paragraphs covering 500 entity types, to facilitate research towards all of these three goals.[1]

## 1   Introduction

Identification and extraction of typed entities in text (Named Entity Recognition, "NER") is a central task in natural language understanding, which is both useful in and of itself and as a central component in other language understanding tasks. Indeed,

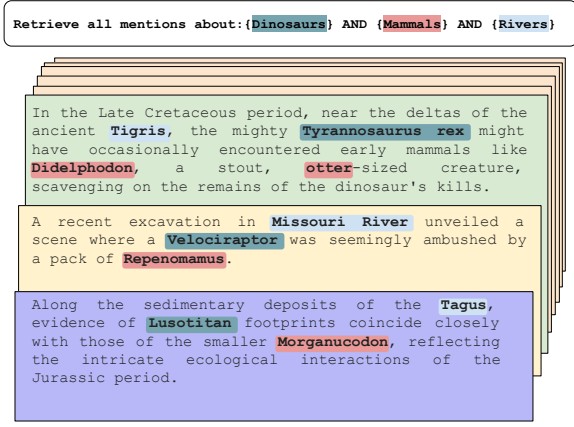

Figure 1: An example of the NER retrieval process for multiple entity types. The retrieval system utilizes entity recognition to retrieve any text that mentions entities from all queried entity types.

it is arguably one of the few pre-Large Language Models natural language technologies that gained wide popularity also outside the NLP research community, and which is routinely used in NLP applications. The NER task has seen large improvements over the past years. Approaches based on fine-tuning of pre-trained models like BERT, RoBERTa, and T5 significantly improve the accuracy of supervised NER models (Devlin et al., 2019; Lee et al., 2020; Phan et al., 2021; Raman et al., 2022), while instruct-tuned LLMs (Ouyang et al., 2022; Chung et al., 2022) seem to offer zero-shot NER capabilities ("What are all the animal names mentioned in this text? {text}"). Moreover, LLMs demonstrate that many language understanding tasks can be performed end-to-end while bypassing the need for a dedicated entity identification stage. Is entity identification and extraction solved or made redundant by pre-trained-based models and LLMs? Is it the end of the research on the NER task? We argue that this is not the end of NER research, but rather a new beginning. The phase transition in abilities enables us to expand our expectations from named-entity

---

[*]Both authors contributed equally to this work.

[1]NERetrieve dataset available at https://github.com/katzurik/NERetrieve

recognition models and redefine the boundaries of the task.

After motivating the need for explicit modeling of named entity identification also in a world with strong end-to-end models §2, we highlight our desiderata for the next-generation entity identification modules §3. These are based on deficiencies we identify with current approaches and abilities §7:

1. Supervised models work well on the identification of broad and general categories ("Person", "Animal"), but still struggle on more nuanced or specialized entity types ("Politician", "Insect"), as well as on intersection types ("French Politicians").

2. While LLM-based models exhibit some form of zero-shot NER, this ability degrades quickly when faced with more fine-grained and specialized types. Both supervised and zero-shot setups fail when applied out-of-domain, identifying many non-entities as entities.

3. While zero-shot NER abilities allow us to identify entities in a given text, it does not allow us to efficiently retrieve all texts that mention entities of this type ("fetch me all paragraphs mentioning an insect").

We thus spell out our vision of next-generation NER: named entity identification which is:

1. Robust across domains.

2. Can deal with fine-grained, specific, and intersectional entity types.

3. Support a zero-shot setup in order to accommodate novel entity types.

4. Extends the zero-shot setup from *identification* to *retrieval*, allowing to index a set of documents such that it efficiently supports retrieval of all mentions of entities of a given zero-shot type. This retrieval should be exhaustive, focusing on the retrieval of all relevant mentions in the collection, and not only the top ones.

We formally define these in the form of NER tasks (§3), and present a silver-annotated dataset that supports research towards these directions (§5).

## 2 Named Entities

The identification of entity mentions is a well-established and central component in language processing, where the input is a text, and the goal is to identify all spans in the text that denote some category of entities. Entity classes can be as broad and generic (PERSON, LOCATION, ORGANIZATION), or specialized to a domain (GENE, PROTEIN, DISEASE), and they may range in specificity (ANIMAL, MAMMAL, DOG_BREED, ...). The predominant setup is the supervised one, in which the set of entity types is given in advance, and a specialized model is trained to perform the task for the given set of types.

Perfectly solving the task requires a combination of strategies, relying on text-internal cues together with external knowledge. For some entity types (e.g., MACHINE LEARNING RESEARCHER) it is common for the text to not provide sufficient hints regarding the type of the mention, and the system should rely on its internal knowledge of the category (a mental list of ML researchers) in order to identify strings of this category. Yet, also in these cases, the system should rely on the text to disambiguate when possible, e.g. mentions of "Michael Jordan" in a text about sports should drive the system against the MACHINE LEARNING RESEARCHER hypothesis, even if its knowledge lists "Michael Jordan" as one. In other cases, the category membership can be inferred based on cues in the surrounding text (mentioning of an entity *barking* suggests its a DOG), or in the string itself (entities of type EDUCATIONAL INSTITUTION often contain words such as *University*, *College* or *School*).

**Named entities in the era of LLMs** LLMs demonstrate increasing levels of ability to perform "end-to-end" language tasks rather than relying on pipeline components. However, we argue that the identification and extraction of entities from text is often useful in and of itself, and not merely as a component in larger tasks like summarization or question-answering (unless the question is "which birds are mentioned in this text", in which case it reduces to NER).

This is most pronounced when we consider the retrieval case, in which a user may be interested in all documents that contain a specific entity ("all news items from 2010-2020 mentioning a French Politician and a Butterfly"), but is relevant also

in the identification case where a user might be interested in scanning a stream and identifying all mentions of a given type. NER-related tasks remain highly relevant also in the age of LLMs.

**Fine-grained, zero-shot NER retrieval as a building block** While the ability the identify entities of a particular type is useful in and of itself, it becomes significantly more useful when used as a building block for an information-seeking application, in which we look for texts that contain several properties. For example, consider the scenario in Figure 1 where a domain expert seeks to analyze the interplay between dinosaurs, mammals, and the river ecosystems they inhabit. By utilizing the NER retrieval system, the researcher can extract passages containing entities from all three specified types, which they could later further refine. This results in a refined dataset, illuminating the coexistence and interactions of these entities in ancient fluvial environments. Such a fine-grained, zero-shot approach revolutionizes how experts sieve through vast scientific literature, ensuring comprehensive insights.

## 3 Next-Generation NER

We now elaborate a sequence of increasingly more challenging criteria for an ideal NER engine, which we believe are now within reach, yet far from being achieved at present and require further research.

While each intermediate ability is useful on its own, when combined they culminate in what we consider to be the holy grail: Transforming NER from Named Entity **Recognition** to Named Entity **Retrieval**.

### 3.1 Cross-domain Robustness

While current supervised NER systems perform well on established benchmarks (Yamada et al., 2020; Zhou and Chen, 2021), they are in fact rather brittle when applied broadly. For example, a system trained to identify CHEMICAL names, when applied to texts (even scientific) that do not mention any chemicals, will still (erroneously) recognize many entities as belonging to the chemical class. We quantify this tendency in Section 7. The next-generation NER system should work also when applied broadly.

**Formal Requirement:** To facilitate this, evaluation for a NER system for a given entity type should

be performed on a wide range of texts that *do not* contain entities of that type.

### 3.2 Finer-Grained, Hierarchical and Intersectional categories

The world of entities is wide and diverse. Over the years, academic supervised-NER research tended towards the recognition of a narrow and restricted set of coarse-grained entity types, such as PERSON, ORGANIZATION, LOCATION (Tjong Kim Sang, 2002) or domain specialized types such as GENE, CHEMICAL, BODY-PART, and so on in the scientific domain (Settles, 2004). Recent datasets aim to extend the set of categories: the OntoNotes corpus features 11 "general purpose" coarse-grain categories (Weischedel et al., 2013), and Few-NERD (Ding et al., 2021) has 8 coarse-grain categories, which are then subdivided into 66 fine-grained ones. However, even Few-NERD's fine-grained categories are rather coarse-grained, taking the coarse-grained category and going one step further down the hierarchy, e.g. from PERSON to POLITICIAN, from BUILDING to AIRPORT, from LOCATION to MOUNTAIN. The world of entity types is vastly richer and broader: categories can be deeply nested into *hierarchies* (ANIMAL → INVERTEBRATE → INSECT → BUTTERFLY), and arranged into *intersectional categories* that combine different properties of the entity *intersectional* (FRENCH POLITICIAN, MIDDLE EASTERN VEGETARIAN DISHES).

Despite significant progress in supervised NER, it is evident that NER systems, even those based on pre-trained language models, still struggle to grasp the underlying nuances associated with entity types (Fu et al., 2020), particularly when considering fine-grained categories (Ding et al., 2021; Malmasi et al., 2022). Already with Few-NERD's conservative fine-grained categories, we observe low supervised NER scores (F1 well below 60%) for many entity types. Hierarchical and Intersectional entity types further impose additional challenges, such as scarcity of data, the need to detect entities from a very long tail which often requires knowledge external to the text, and the need to make fine-grained distinctions between closely related types. In section §7.2 we demonstrate the deterioration of supervised NER accuracy as we move towards deeper levels of hierarchy and towards intersections.

As LLMs seem to excel at gathering vast and relevant world knowledge in their pre-training stage and using it in their predictions, this raises hopes

towards accurate entity recognition systems that can handle the rich set of possible entity types, at various levels of granularity.

**Task definition: Fine-grained Supervised NER** The supervised NER task is well established in its form: a training stage is performed over texts annotated with typed, labeled spans, where the types come from a predefined set of types denoting entities. At test time, the system gets as input new, unlabeled texts, on which it should identify and mark labeled spans of the entity types seen during training. Future supervised NER systems should retain the established task definition, but be trained and tested on a much richer set of entity types, which includes a wide range of fine-grained, hierarchical, and intersectional entity types. Ideally, we should aim for the training set to be as small as possible.

### 3.3 From supervised to zero-shot based NER

While supervised NER is effective, it requires the collection of a dedicated training set. Users would like to identify entities of their novel types of interest, without needing to collect data and train a model. This desire becomes more prominent as the range of entity types to identify grows and becomes more granular: constructing supervised NER systems for each category becomes impractical. We thus seek a zero-shot system, that can identify entity types not seen in training, based on their name or description.

The zero-shot task is substantially more challenging than the supervised one, requiring the model to rely on its previous "knowledge" to both scope the boundaries of the type to be identified given a class name or a description, as well as to identify potential members of this class (e.g. knowing that Lusotitan atalaiensis is a dinosaur) and identifying supporting hints for class membership (if someone obtained a Ph.D. from X then X is an educational institute).

Recent experience with LLMs suggests that, for many instances, the model indeed possesses the required knowledge, suggesting the zero-shot setup is achievable (Wang et al., 2022b; Agrawal et al., 2022; Hu et al., 2023; Wei et al., 2023).

**Task definition: Zero-shot Fine-grained NER** In the zero-shot setup, the model can be trained however its designer sees fit. At test time, the model is presented with a text, and an entity category name ("dog breeds"), and should mark on

the text all and only spans that mention an entity of this type. Unlike the supervised case, here the desired entity type is not known at train time, but rather provided to the model as an additional input at test time.

Like in the supervised case, the ideal zero-shot model will be able to support fine-grained, hierarchical, and intersectional types.

### 3.4 From entity identification to exhaustive retrieval

Traditional NER tasks focus on the *identification* (or *recognition*) of entities within a text. This requires a model to *process the text* for any entity of interest, which can become both slow and expensive when users want to extract mentions of arbitrary entity types from a large text collection, where most documents do not contain an entity of interest ("what are all the snakes mentioned in this corpus").

We thus propose the challenge of **NERetrieve**: moving the NER task from zero-shot *recognition* to zero-shot *retrieval*. In this setup, a model is used to process a large corpus and index it, such that, at test time, a user can issue entity-type queries for novel entity types, and the model will return *all* mentions of entities of this type in the collection (like in the zero-shot setup), but without scanning all the documents in the collection.

Crucially, unlike standard retrieval setups which are focused on finding the *most relevant* documents in a collection, the NERetrieve setup is **exhaustive**: we aim to locate *all* the relevant documents in the indexed collection.

**Task definition: Exhaustive Typed-Entity Retrieval** In the exhaustive retrieval setup, the entity type label (such as "DOG BREEDS") functions as a query to retrieve a comprehensive collection of all relevant documents containing mentions of entities belonging to that specific category. During testing, the model will encounter previously unseen queries (entity types) and will be required to retrieve all available documents and mark the relevant spans in each one.

### 3.5 To Summarize

Supervised NER is still not "solved" for the general case, and future techniques should work towards working also with fine-grained and specialized entity types, and being robust also on texts that do not naturally mention the desired entity types. A

step further will move from the supervised case to a zero-shot one, where the entity type to be identified is not known at train time, but given as input at test time. Evidence from LLMs suggests that such a task is feasible, yet still far from working well in practice. Finally, we envision moving zero-shot NER from *recognition* to *retrieval*, in a setup that assumes a single pre-processing step over the data, indexing, and allowing to retrieve all mentions of entities of a given category which is unknown at indexing time and only specified at query time.

## 4 Relation to Existing Tasks

**Fine-grained NER**    While the "canonical" NER task in general-domain NLP centers around very broad categories (PERSON, LOCATION, ORGANI-ZATION, ...), there have been numerous attempts to define more fine-grained types, often by refining the coarse-grained ones. Sekine and colleagues (Sekine et al., 2002; Sekine, 2008) converged on a schema consisting of 200 diverse types, and Ling and Weld (2012) introduced FIGER, a Freebase-based schema based on the Freebase knowledge graph, comprising 112 entity types. Ding et al. (2021) was inspired by FIGER schema when annotating "Few-Nerd", which consists of 66 fine-grained entity types and is the largest fine-grained NER dataset that was annotated by humans. In contrast to the schema-based sets that aim to define a universal category set that will cover "everything of interest", we argue that such a set is not realistic, and each user has their own unique entity identification needs. Hence, the label set should be not only fine-grained but also dynamic, letting each user express their unique needs. Our vision for NER is thus *zero-shot over very fine-grained types* and our selection of 500 categories is thus not supposed to reflect a coherent "all-encompassing" set, but rather an eclectic set of fine-grained and often niche categories, representative of such niche information needs. Entity typing, a closely related field to NER, aligns with our proposed vision.

Choi et al. (2018) introduced the concept of Ultra-Fine Entity Typing: using free-form phrases to describe suitable types for a given target entity. While this echoes our interest in ultra-fine-grained types, in their task the model is presented with an in-context mention and has to predict category-names for it. In contrast, in the zero-shot NER tasks, the user gets to decide on the name, and the model should recognize it. Nonetheless, approaches to the entity-typing task could be parts of a solution for the various NER tasks we propose.

**Few-shot NER**    Work on Few-shot NER (Wang et al., 2022b; Ma et al., 2022; Das et al., 2022) assumes a variation on the supervised setup where the number of training samples is very small (below 20 samples). While this setup indeed facilitates schema-free NER, we argue that it still requires significant effort from the user, and that a zero-shot approach (which given LLMs seems within reach) is highly preferable. Our vision for NER is zero-shot, not few-shot. Our dataset, however, can also easily support the few-shot setup for researchers who choose to tackle a challenging, fine-grained variant of it. Compared to FewNERD (Ding et al., 2021), the current de-facto standard for this task, we provide more and more fine-grained entity types.

**Zero-shot NER**    Zero shot NER is already gaining momentum (Hu et al., 2023; Wang et al., 2023). We support this trend, and push towards a finer-grained and larger scale evaluation.

**Entity Retrieval and Multi-Evidence QA**    Entity-retrieval tasks aim to retrieve information about entities based on a description. For example, Hasibi et al. (2017) introduced the LISTSEARCH subtask in DBpedia-Entity v2, in which the user enters fine-grained queries with the aim of retrieving a comprehensive list of DBpedia matching the query, and Malaviya et al. (2023) introduced QUEST, a dataset for entity retrieval from Wikipedia with a diverse set of queries that feature intersectionality and negation operations, mirroring our desire for finer-grained and intersectional types. In (Amouyal et al., 2022; Zhong et al., 2022), the set of entities to be retrieved is defined based on a question that has multiple answers ("Who is or was a member of the Australian Army?").

These tasks all focus on retrieving *a set of entities* based on a description. In contrast to this type-focused view, the **NERetrieve** task is concerned instead with *mentions within documents* and requires finding not only the set of entities but also *all mentions* of these entities in a collection of text. This includes complications such as resolving ambiguous mentions, finding aliases of entities, and dealing with ambiguous strings that may or may not correspond to the entity.

These entity-retrieval tasks described in this section can be an initial component in a system that attempts to perform the complete **NERetrieve** task.

## 5 The NERetrieve Dataset

Having spelled out our desired future for NER tasks, we devise a dataset that will facilitate both research and evaluation towards the range of proposed tasks, from fine-grained supervised recognition to zero-shot exhaustive retrieval.

Firstly, to support **supervised fine-grained, hierarchical and intersectional Named Entity Recognition (NER)**, the dataset is designed to contain a large number of entity types (500) selected such that they include a wide array of fine-grained, domain-specific, and intersectional entity types, as well as some easier ones. Each entity type encompasses a substantial number of entities and entity mentions, which are scattered across thousands of distinct paragraphs for each entity type, thereby providing a rich, varied resource for effective supervised fine-grained NER tasks, and allowing evaluation also on specific phenomena (e.g., only hierarchical types, only intersectional, and so on). The large number of classes and paragraphs also provides a good testbed for robustness, verifying that a NER system trained on a subset of the classes does not pick up mentions from other classes.

The same setup transfers naturally also to the **Zero-shot fine-grained NER task**, as the entity types collection is designed to reflect a diverse range of interests and world knowledge, similar to what potentially causal users would expect from a zero-shot NER system. We also provide an *entity-split*, detailing which entity types can be trained on, and which are reserved for test time.

Finally, to support the novel **exhaustive typed-entity retrieval task**, the dataset is tailored to suit scenarios where knowledge corpora are massive, encompassing millions of texts. This necessitates more complex methodologies than those currently available, and our dataset is specifically constructed to facilitate the evolution of such advanced techniques. Each entity type potentially corresponds to thousands of different paragraphs with a median of 23,000 relevant documents for each entity type. Each entity type label serves as a descriptive name that can be used as a query. At the same time, we kept the dataset size manageable (4.29M entries) to allow usage also on modest hardware.

A final requirement is to make the dataset **free to use and distribute**, requiring its construction over an open resource.

**We introduce NERetrieve**, a dataset consisting of around 4.29 million paragraphs from English Wikipedia. For each paragraph, we mark the spans of entities that appear in it, from a set of 500 entity types. The paragraphs were selected to favor cases with multiple entity types in the same paragraph. The selection of entity types was carefully curated to include a diverse range of characteristics. This includes a combination of low and high-level hierarchy (Insects and Animal), First and second-order intersections (Canoeist and German Canoeist), easier (Company) and more challenging types (Christmas albums), as well as types with both few and numerous unique entities. This approach ensures a balanced representation of different entity characteristics in the dataset. Due to the increasing specificity in entity types, each corpus span might be marked for multiple entity types.

We provide an 80%/20% train/test split, ensuring that each split maintains an approximate 80%/20% distribution of all entity types (for the supervised NER setup). Furthermore, paragraphs are unique to each set and do not overlap. In addition, for the zero-shot NER and the retrieval setups, we designate 400 entity types for training and 100 for testing.[2]

## 6 Technical Details of Dataset Creation

With more than 4 million paragraphs and 500 entity types, exhaustive human annotation is not feasible. We therefore seek an effective method for creating silver-annotated data, while retaining high levels of both precision (if a span is marked with a given type, it indeed belongs to that type) and comprehensiveness (all spans mentioning a type are annotated). At a high level, we rely on high-quality human-curated entity lists for each entity type (part of the inclusion criteria for an entity type is the quality of its list), where each list includes several aliases for each entity. We then collect paragraphs that include relaxed matches of the entities in our entity lists.[3] Due to language ambiguity, the resulting matches are noisy, and some marked spans do not refer to the desired entity (not all mentions of "the Matrix" are of the science fiction movie). We filter these based on a contextual classifier trained over a very large document collection, which we find to perform adequately for this purpose.

---

[2]See Appendix A.1 for examples

[3]While no list is fully comprehensive, taking only paragraphs on which entities from the list matched to begin with reduces the chances of the texts to include a non-covered entity, compared to using random texts and annotating them.

**Curating Entity Lists** Each entity type in our dataset corresponds to an ontology from the Caligraph knowledge base (Heist and Paulheim, 2020)[4] which is in turn based on a combination of DBPedia Ontology, and Wikipedia categories and list pages. We collect 500 ontologies encompassing a total of 1.4M unique entities[5], and we further augment each entity with all possible aliases in the corresponding Wikidata[6] entity page under the "Also known as" field.

We selected ontologies such that the resulting set covers a diversity of topics, while also ensuring that each individual ontology has high coverage, and represents a concrete meaningful category, excluding abstract or time-dependent categories. Certain ontologies were identified as subsets of larger ontologies, thereby establishing a hierarchical structure among the types (e.g., the "Invertebrate" ontology is part of the broader "Animals" ontology). We also ensured that many, but not all ontologies are of intersectional types.

**Tagging and Filtering Entity Mentions** Each entity within the dataset was mapped to all Wikipedia articles that referenced it, based on DBpedia's link graph (Lehmann et al., 2015). As a result, the extraction of entities from Wikipedia article texts occurred solely on Wikipedia pages where at least one of the entities was explicitly mentioned. We indexed this document set with ElasticSearch, which we then queried for occurrences of any of the 1.4M entities or their aliases, using a relaxed matching criterion that requires all the entity words to match exactly, but allows up to 5 tokens (slop 5) between words.

The result of this stage matched significantly more than 4.29M paragraphs, and is exhaustive but noisy: not all spans matching an alias of a typed entity are indeed mentions of that entity. We resolved this using a contextual filtering model, which we found to be effective. We then apply this model to select paragraphs that both contain many entities of interest, *and* were determined by the filtering model to be reliable contexts for these entities.

Our filtering model is a Bag-of-Words Linear SVM, trained individually for each entity type using TF/IDF word features. Positive examples are from a held-out set matching the type, while nega-

tives are randomly chosen paragraphs from other entity types. The model essentially determines the likelihood of encountering entities of type X within a given text. The choice of a linear model reduces the risk of entity-specific memorization (Tänzer et al., 2022), thereby focusing on the context instead of specific entities. Preliminary experiments indicate its efficiency at scale. The model is run on all paragraphs for each entity, discarding those predicted as low compatibility with the respective entity type.

**Quality Assessment of the Resulting Dataset** To assess the reliability of the proposed dataset, we randomly selected 150 paragraphs and manually evaluated each tagged entity and its corresponding entity types. This process consisted of two steps: firstly, confirming the accurate tagging of entities within the text, and secondly, ensuring the correctness of each `{entity, entity type}` pair. The 150 paragraphs collectively contained 911 typed entity mentions, with each typed mention being cross-validated against Wikipedia, which serves as the source for both the entity types and the paragraphs. Of these 911 typed mentions, 94% were tagged accurately. While not perfect, we consider this to be an acceptable level of accuracy considering the datasets size, coverage, and intended use. Nevertheless, we plan to release the dataset together with a versioning and reports mechanism that will allow future users to submit mistakes, resulting in more accurate versions of the dataset over time.

## 7 Performance of Current Models

The **NERetrieve** dataset allow us to assess the performance of current models on the challenging conditions we describe in section 3.

**Models** For the supervised case, we train (fine-tune) Spacy[7] NER models over a pre-trained DeBERTa-v3-large MLM. For the zero-shot scenarios, we prompt gpt-3.5-turbo and GPT4.

### 7.1 Robustness on broad-scale texts

The first experiment quantifies the robustness of NER models when applied to domain-adjacent texts, texts that do not contain and are not expected to contain, entities of the types the model aims to identify.

For the *supervised case*, we selected 20 random entity types. For each entity type, we trained a

---

[4] `caligraph.org`
[5] See Appendix A.3 for entity types
[6] `wikidata.org`

[7] `https://spacy.io/`

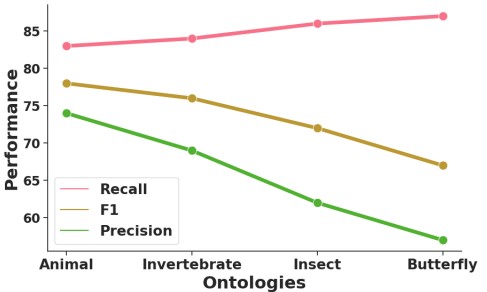

Figure 2: NER performance across increasing levels of granular hierarchical classification. Model precision decreases as the entity type becomes more fine-grained, owing to the misclassification of similar types

| | Exact | | | Relaxed | | |
|---|---|---|---|---|---|---|
| Model | P | R | $F_1$ | P | R | $F_1$ |
| GPT3.5 | 0.23 | 0.27 | 0.22 | 0.32 | 0.37 | 0.31 |
| GPT4 | 0.38 | 0.56 | 0.41 | 0.53 | 0.75 | 0.57 |

Table 1: Zero-shot NER Performance of GPT-3.5 and GPT-4 on 1,000 Paragraphs from NERetrieve. Paragraphs were sampled using a stratified method to ensure a uniform distribution of each entity type.

NER model with a train set of 15,000 samples that included that specific entity type. We then tested this model on a separate set of 10,000 randomly selected paragraphs that did not have the targeted entity type. On average, models incorrectly identified an entity (a False Positive) in **1 out of every 4** paragraphs. After manually reviewing a portion of these false positives, we confirmed that none of them actually contained the specified entity type. This stresses the brittleness of supervised models in this setting. For the *Zero-shot* setting, a prompted gpt-3.5-turbo model was remarkably more robust, only identifying a false positive in **1 out of 500** paragraphs. This highlights the robustness of the LLMs in the zero-shot setting and suggests that such robustness should also be possible for dedicated, supervised models.

### 7.2 Supervised Fine-grained NER

**Increasing level of specificity** We measure existing supervised and zero-shot NER performance, as we increase the specificity of the type to be identified. We consider a hierarchical chain of entity types wherein each subsequent type has greater granularity levels than the previous one: ANIMAL → INVERTEBRATE → INSECT → BUTTERFLY. We train a supervised NER model for each type. Each training set consisted of 15,000 paragraphs, and evaluate all of them on all the paragraphs that contain the ANIMAL type (which also includes the other types).

The results in Figure 2 confirm that both precision and F1 indeed drop significantly as we increase the level of specificity.

**Intersection-types** Intersection types require the model to accurately distinguish closely related entities with fine-grained distinctions. To demonstrate the challenge, we selected five related intersected entity types: INDIAN WRITER, NORTH AMERICAN WRITER, BRITISH WRITER, POLISH WRITER, and NORWEGIAN WRITER. Here, we trained a 5-way NER model on a train-set containing entities of all these types. The model was evaluated on a test set containing these types as well as related entities not present in the train set, which consists of professions such as AMBASSADOR, JUDGE, DESIGNER, PAINTER, and GUITARIST. This was done to assess the model's ability to differentiate between Writers and other professions and to differentiate authors of different nationalities. The model achieved a very low micro-average **F1 score of 0.14 (precision: 0.08, recall: 0.73)**, demonstrating the difficulty of the task. We then combined all five writer-types above to a single Writer class. A NER model trained for this class has an **F1 score of 0.44 (precision: 0.31, recall: 0.79)**, substantially higher but still very low.

### 7.3 Zero-Shot Fine-Grained NER with LLM

To assess current LLMs ability to perform zero-shot NER over a wide range of fine-grained entity types, we designed the following experiment. We randomly draw 1,000 paragraphs from the dataset, while attempting to achieve an approximately uniform distribution of entity types in the sample. For each paragraph, we prompt the LLM with the paragraph text, and asked it to list all entities of types that appear in the paragraph, as well as an additional, random entity type (for example, for a paragraph containing mentions of FOREST and PROTECTED AREA the prompt would ask to extract mentions of FOREST, PROTECTED AREA and LIGHTHOUSE[8] We assess both GPT-3.5-turbo (Ouyang et al., 2022) and GPT-4 (OpenAI, 2023), using their function calling API features to structure the output in JSON format.

---

[8]See Appendix A.2 For prompt and examples.

| Model | Recall@\|REL\| |
|---|---|
| BM25 | $0.221 \pm 0.16$ |
| E5-v2-base | $0.331 \pm 0.15$ |
| E5-v2-large | $0.326 \pm 0.14$ |
| GTE-base | $0.396 \pm 0.16$ |
| GTE-large | $\mathbf{0.397 \pm 0.16}$ |
| BGE-base | $0.383 \pm 0.16$ |
| BGE-large | $0.369 \pm 0.14$ |

Table 2: Performance evaluation of Exhaustive Typed-Entity Mention Retrieval on the test set in a Zero-Shot setup: mean and standard deviation of Recall@\|REL\| across all entity types.

In Table 1 We report the Recall, Precision, and F1 scores for both the Exact Match, wherein the generated string is compared directly with the tagged entity in the dataset, and the Relaxed Match, which considers a match based on word overlap between the generated and the gold entity in the dataset. Notably, we observed a substantial improvement in all metrics with GPT-4 compared to GPT-3.5-turbo. However, despite these advances, the results remain relatively low, indicating a considerable potential for further improvement in this task. These findings are consistent with other studies that highlight the struggles encountered by GPT models in zero-shot NER tasks across classical NER tasks (Wang et al., 2023). This challenge becomes even more pronounced when the entity types become more fine-grained and specific, resulting in a performance that falls short when compared to supervised models (Hu et al., 2023).

### 7.4 Exhaustive Typed-Entity Mention Retrieval

To our knowledge, no current system is directly applicable to our proposed **NERetrieve** task: text embedding and similarity-based methods are not designed for the entity-type identification task, and retrieval systems are not designed to support exhaustive retrieval of all items (each of our queries should return a median of 23,000 paragraphs) but rather focus on returning the top-matching results. Indeed, this task stresses the limits of both retrieval and semantic embedding methods, calling for future research. We nonetheless make a best-effort experiment within the existing models and methods.

We evaluate NER-retrieval on BM25 (Robertson et al., 2009) for sparse retrieval, and on three state-of-the-art sentence encoder models for dense retrieval: BGE (Xiao et al., 2023), GTE (Li et al.,

2023) and E5-v2 (Wang et al., 2022a). The dense models were selected as they held the leading position in the retrieval category of MTEB text embedding benchmark (Muennighoff et al., 2023)[9], and exhibit particularly strong performance in the DBpedia-entity-v2 task (Hasibi et al., 2017), a specific IR task designed for entity retrieval. Additionally, for the sentence encoder models, we encoded each paragraph in the dataset as a vector and measured cosine similarity with the query vector to determine the ranking. We report the performance of the IR systems with **Recall@\|REL\|** metric, where \|REL\| represents the total number of relevant documents associated with a given query. This metric is analogous to the R-Precision measurement (Sanderson, 2010). We compute the average of this metric over each of the 100 entity types in the test set, where the entity type serves as the query. Based on the findings in Table 2, it becomes evident that despite utilizing state-of-the-art models for semantic search, our progress in this task is significantly distant from our desired objectives. Intriguingly, the number of relevant documents within the entity type does not demonstrate a correlation[10] with the Recall@\|REL\| measurement, for the E5-v2-base experiment. This suggests that the task is not challenging solely due to the existence of a vast volume of texts, but rather because of the inherent complexity of the task itself.

## 8 Conclusion

We argued that NER remains a relevant and challenging task even in the age of LLMs: rather than solving NER, the introduction of LLMs allows us to take the NER task to the next level.

We highlight three challenging directions in which the task can evolve, culminating in a move from named entity recognition to named entity retrieval, and created a unique dataset that allows future explorations of these directions.

## 9 Limitations

**Silver Data** The dataset we provide is silver-annotated, meaning it's automatically annotated by an algorithm and not manually curated. While this allows for scalability and provides a large number of examples, it inevitably introduces errors and inconsistencies. Silver annotations, while useful

---

[9]`huggingface.co/spaces/mteb/leaderboard`

[10]Pearson correlation coefficient $r = -0.06$

for training models in the absence of extensive manually-annotated gold-standard datasets, are not a perfect substitute. Their accuracy depends on the performance of the algorithms used for annotation, and there may be variations in quality across different types of entities or texts.

**Exhaustiveness** In our pursuit of constructing a comprehensive corpus, it is essential to acknowledge potential limitations inherent to our dataset. We recognize the possibility of underrepresentation for certain entity types or categories, particularly those that are rare or lack a dedicated Wikipedia article. Furthermore, nuanced details may occasionally go unnoticed. Nevertheless, our resource comprises an extensive catalog of 1.4 million unique entities, indicating a significant stride toward comprehensive data coverage. It presents a notable attempt to achieve a level of exhaustiveness within the bounds of feasibility while striving to exceed the limitations inherent in manual human annotation efforts.

While these limitations should be considered, they do not diminish the potential of our resources and our vision to stimulate innovation in NER research. We encourage the research community to join us in refining this resource, tackling the highlighted challenges, and moving toward the next generation of Named Entity Recognition.

## 10 Ethics and Broader Impact

This paper is submitted in the wake of a tragic terrorist attack perpetrated by Hamas, which has left our nation profoundly devastated. On October 7, 2023, thousands of Palestinian terrorists infiltrated the Israeli border, launching a brutal assault on 22 Israeli villages. They methodically moved from home to home brutally torturing and murdering more than a 1,400 innocent lives, spanning from infants to the elderly. In addition to this horrifying loss of life, hundreds of civilians were abducted and taken to Gaza. The families of these abductees have been left in agonizing uncertainty, as no information, not even the status of their loved ones, has been disclosed by Hamas.

The heinous acts committed during this attack, which include acts such as shootings, raping, burnings, and beheadings, are beyond any justification.

In addition to the loss we suffered as a nation and as human beings due to this violence, many of us feel abandoned and betrayed by members of our research community who did not reach out and were even reluctant to publicly acknowledge the inhumanity and total immorality of these acts.

We fervently call for the immediate release of all those who have been taken hostage and urge the academic community to unite in condemnation of these unspeakable atrocities committed by Hamas, who claim to be acting in the name of the Palestinian people. We call all to join us in advocating for the prompt and safe return of the abductees, as we stand together in the pursuit of justice and peace.

This paper was finalized in the wake of these events, under great stress while we grieve and mourn. It may contain subtle errors.

## Acknowledgements

We would like to thank Nicolas Heist of the Caligraph project for his assistance and feedback.
This project has received funding from the European Research Council (ERC) under the European Union's Horizon 2020 research and innovation programme, grant agreement No. 802774 (iEXTRACT).

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

# A Appendix

## A.1 Example Paragraphs from the NERetrieve Dataset

---

*Canelo Álvarez vs. Julio César Chávez Jr.*[Sports event/Boxer] was a professional boxing fight held on May 6, 2017 at the *T-Mobile Arena*[Building/Stadium] in *Paradise, Nevada*[City/City in the Americas], part of the Las Vegas metropolitan area. *Álvarez*[Sportsman/Boxer] was declared the winner by unanimous decision, having been judged the winner of all 12 rounds by each of the three ringside judges

---

*Aristotelia devexella*[Arthropod/Insect/Animal/Moth/Eukaryote/Invertebrate/Specie] is a moth of the family Gelechiidae. It was described by *Annette Frances Braun*[Scientist/Entomologist] in 1925. It is found in North America, where it has been recorded from Alberta, Arizona and Oklahoma.

---

In June 1921, *Malacrida*[Poet/British_writer/Women_writer] met her future husband, *Marchese Piero Malacrida de Saint-August*[Journalist/Designer], an Italian journalist and former cavalry officer, at a charitable fundraising event known as Alexandra Rose Day at *The Ritz Hotel, London*[Building/Hotel]. They were married on 6 December 1922, at *St Bartholomew-the-Great*[Building/Church/Monastery], making her the Marchesa Malacrida de Saint-August. Her husband's family were a noble family from Lombardy. Shortly after their wedding, her husband expanded his activities into writing on interior design, and designing interiors, especially luxury bathrooms, for the upper class. The couple would buy flats at smart *London*[City] addresses, then remodel and sell them, trading under the name Olivotti. In 1926-1929, they lived at 4 Upper Brook Street, *Mayfair*[District_of_London].

---

During the intense study of the fossils it became clear that they did not represent *Barosaurus*[Dinosaur] but a species new to science. In 2012 this was named *Kaatedocus siberi*[Dinosaur], by the Swiss palaeontologist *Emanuel Tschopp*[Scientist], who as a boy had visited the excavations, and his Portuguese colleague *Octávio Mateus*[Scientist]. The generic name, which means small beam, combines a reference to the related form *Diplodocus*[Dinosaur] with a Crow Indian diminutive suffix kaate. The specific name honours Siber.

---

In 1995, the Government of Bangladesh declared *Sylhet*[City] as the sixth divisional headquarters of the country. *Sylhet*[City] has played a vital role in the Bangladeshi economy. Several of Bangladesh's finance ministers have been Members of Parliament from the city of *Sylhet*[City]. *Badar Uddin Ahmed Kamran*[Bengali_politician/Mayor] was a longtime mayor of *Sylhet*[City]. *Humayun Rashid Choudhury*[Bengali_politician/Ambassador/Diplomat], a diplomat, served as President of the UN General Assembly and Speaker of the Bangladesh National Parliament.

---

Table 3: Annotated paragraphs from the **NERetrieve** dataset. Entities may correspond to multiple entity types, representing diverse aspects. In this example, different entity types associated with the same entity are separated by slashes.

## A.2 Zero-shot NER with GPT - Prompt and examples

The following JSON is used for the function calling feature in OpenAI API[11]. `entity_types` represents a list of entity type names separated by a comma.

```
{
  "name": "zero_shot_ner",
  "description": f"For each entity type: {entity_types}. Extract all relevant
      entities. Entity can appear in more than one entity type.",
  "parameters": {
    "type": "object",
    "properties": {
      "entities": {
        "type": "array",
        "items": {
          "type": "object",
          "properties": {
            "entity_type": {
              "type": "string",
              "description": "Category of the named entity."
            },
            "entities": {
              "type": "string",
              "description": "A list of Named entities extracted from text."
            }
          }
        }
      }
    }
  },
  "required": ["entities"]
}
```

---

A performance of the complete *organ works of Johann Sebastian Bach*[Classical composition] was held on November 22, 2014 at *St. Peter's Lutheran Church*[Religious leader] in Manhattan. Twenty organists from *The Juilliard School*[School/Educational institution] performed. *Juilliard Organ Department Chair, Paul Jacobs*[Composer/Classical musician], curated the eighteen-hour performance. *Johann Sebastian Bach*[Composer/European musician/Classical musician/German classical musician] was a key figure in the event.

---

*Jimmy Dorsey*[Jazz musician/Saxophonist] is considered one of the most important and influential alto saxophone players of the Big Band and Swing era, and also after that era. Jazz saxophonists *Lester Young*[Jazz musician/Saxophonist] and *Charlie Parker*[Jazz musician/Saxophonist] both acknowledge him as an important influence on their styles.

---

*Lapland Reserve*[Forest] is located in the *Scandinavian and Russian taiga ecoregion*[Forest], which is situated in Northern Europe between tundra in the north and temperate mixed forests in the south. It is covers parts of Norway, Sweden, Finland and the northern part of European Russia, being the largest ecoregion in Europe. The ecoregion is characterized by coniferous forests dominated by *Pinus sylvestris*[Forest], often with an understory of *Juniperus communis, Picea abies and Picea obovata*[Forest] and a significant admixture of *Betula pubescens and Betula pendula*[Forest]. *Larix sibirica*[Forest] is characteristic of the eastern part of the ecoregion.

---

Table 4: Example of GPT4 extracted entities and entity types. Entities may correspond to multiple entity types, representing diverse aspects. False positive entity types are marked in red. In this example, different entity types associated with the same entity are separated by slashes.

---

[11]https://platform.openai.com/docs/guides/gpt/function-calling

## A.3 Entity Types Infographic

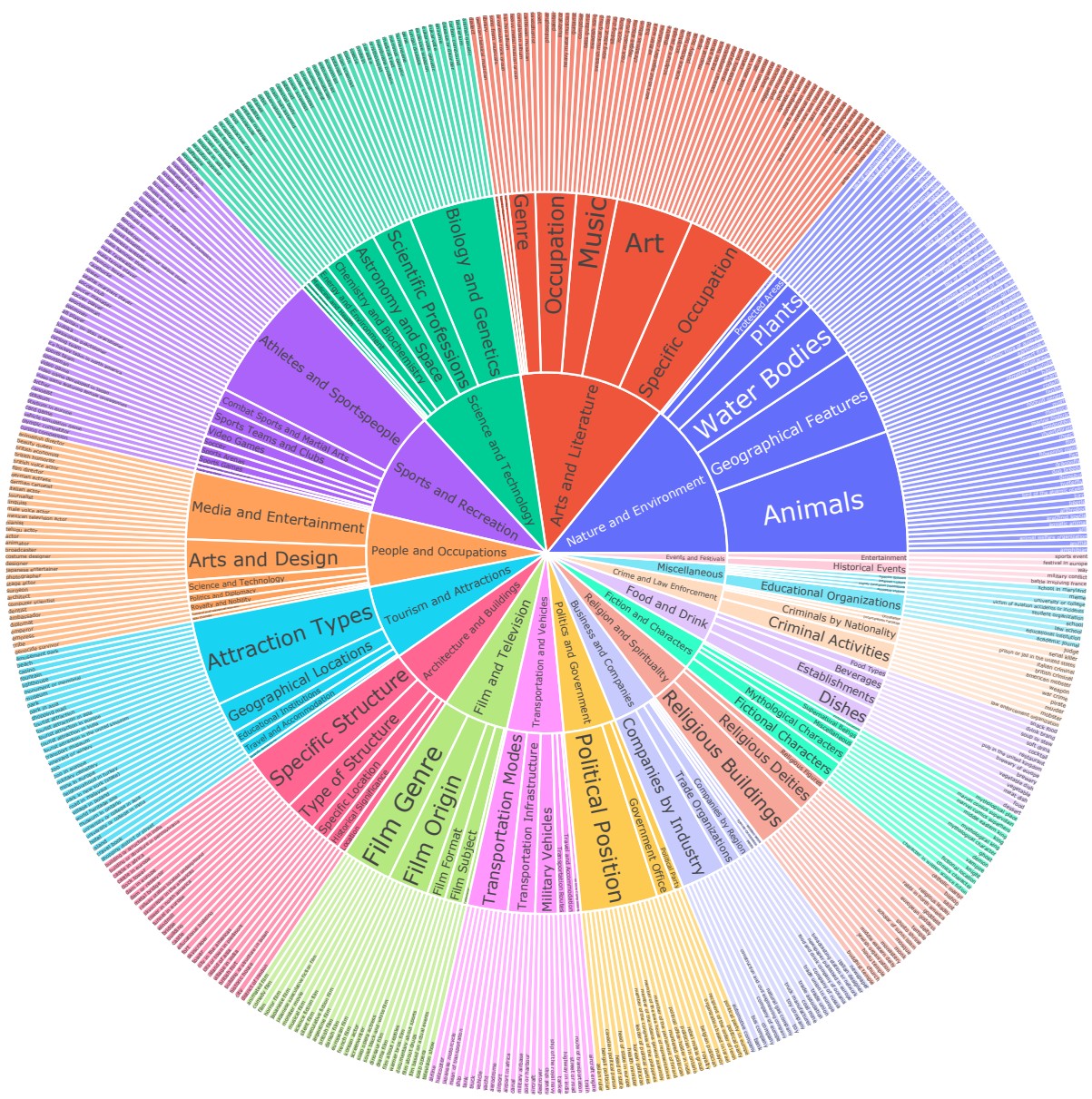

Figure 3: All the 500 entity types in the dataset are arranged topically in an easy-to-skim diagram. Note that this diagram does not reflect all the hierarchical relations between entity types in the dataset, as it is capped at a hierarchy level 3 for aesthetic reasons.