# OpenReview forum: "NERetrieve: Dataset for Next Generation Named Entity Recognition and Retrieval"
_EMNLP/2023/Conference — EMNLP 2023 Findings_

### Official Review · Reviewer_EbDr · 2023-08-01

**Soundness:** 3

**Excitement:**

4: Strong: This paper deepens the understanding of some phenomenon or lowers the barriers to an existing research direction.

**Missing References:**

- see earlier: works on skill extraction also amount to span extraction & classification, and thus seem worthy to refer to; see [1,2] above, as just 2 examples of such works -- there could be others that are better suited to refer to;

[1] https://aclanthology.org/2023.acl-long.662/
[2] https://aclanthology.org/2020.coling-main.513/


**Paper Topic And Main Contributions:**

This is a position paper, carefully assessing the state of current NER solutions, particularly those based on the most recent LLMs. Specifically, the authors define (and establish current models' performance on) three challenging NER tasks: (1) fine-grained supervised NER (with fine-grained, hierarchical and intersectional entity types); (2) zero-shot fine-grained NER (mark all mentions of a stated type, e.g., "dog breed", in a given text); (3) exhaustive typed-entity retrieval (e.g., return all documents that mention a dog breed + the span doing so). Moreover, the authors offer a new dataset (and describe its collection & automated annotation process) to support research & development of new models on aforementioned tasks, NERetrieve (4M paragraphs, having entity spans marked with one of the 494 considered types).

The paper contributions are (1) the definition of challenging NER tasks (including retrieval), (2) a new dataset, (2) performance analysis of recent models using that dataset for the three challenging NER tasks.
The authors conclude that for all of the tasks also the most recent large-scale LLM models do not solve them, thus setting out a path for future research.


**Questions For The Authors:**

- Do you see any link to works on XMLC tasks? E.g. the dataset presented in [2]

[2] https://aclanthology.org/2020.coling-main.513/


**Reasons To Accept:**

- nice description/definition of real-world use cases of challenging NER tasks
- contribution to the broad NLP research community with a sizeable dataset supporting effective analysis of new solutions to the 3 defined challenging NER tasks

**Reasons To Reject:**

- no reference is made to works on (extreme) multi-label classification tasks, e.g., on skill extraction (just as 1 example, see [1] & the multiple references in its introduction section); that line of work also pertains to an extraction task (i.c. skill extraction) with many possible labels that are organized in a hierarchical taxonomy; this is conceptually very much related to fine-grained NER
- the selection of baseline/reference models for the newly defined tasks is not very clearly motivated

[1] https://aclanthology.org/2023.acl-long.662/


**Reproducibility:**

4: Could mostly reproduce the results, but there may be some variation because of sample variance or minor variations in their interpretation of the protocol or method.

**Reviewer Confidence:**

3: Pretty sure, but there's a chance I missed something. Although I have a good feel for this area in general, I did not carefully check the paper's details, e.g., the math, experimental design, or novelty.

**Typos Grammar Style And Presentation Improvements:**

- Please position footnote markers (superscript numbers) *after* punctuation. (E.g., at the end of the Abstract)
- Use curly quotes please, rather than straight ones (e.g,. 072 and following)
- Always use a comma after "e.g."
- 067,069,071:  put quotes around references to other sections:  §2 -> (§2)  etc.
- 119: e.g, -> e.g.,   [missing '.']
- 150:  Buttefly -> butterfly   [no capital; missing 'r']
- 162-63: join 2 citations within same pair of brackets?
- 169: Section 7 -> §7   [be consistent; elsewhere you also simply use '§']
- 187: coarse-grain -> coarse-grained   [be consistent, please]
- 268: missing space after 'achievable'
- 300-301: "without scanning all the documents":  what does this mean? How can the result be "exhaustive" if there is a priori a filter that excludes some documents to be considered?
- 372: assume -> assumes   [subject is 'work' on line 370]
- 589: why did you pick DeBERTa, and not some other BERT variant? Please add a motivation
- 603: "s.d. 0.21":  what does s.d. mean? standard deviation? on what quantity is it the standard deviation exactly?
- 604-5: and none were indeed -> and indeed, none were
- 611: suggesting -> suggests
- 658 etc.: "attempting to achieve an approximately uniform distribution of entity types":  please clarify how your approach would indeed amount to a uniform selection over all possible types. Given that you randomly draw paragraphs, wouldn't the distribution of selected types rather follow their (non-uniform?) distribution in terms of number of mentioned instances?
- 693: NERertiev -> NERetrieve
- Table 2: "caption caption caption"?
- 722: "analogous": vague/imprecise; either it is exactly the same or not?

---

> ### Author Rebuttal · Authors · 2023-08-29
>
> **Thank you for recognizing our detailed portrayal of real-world NER challenges and our dataset's contribution to advancing NLP research.**
>
> Please see our response below:
>
> **P1.** *No reference is made to works on (extreme) multi-label classification tasks..*
>
> Thank you for pointing out this direction. We see its relevance and will incorporate appropriate references in the camera-ready version.
>
> **P2.** *The selection of baseline/reference models for the newly defined tasks is not very clearly motivated*
>
> We appreciate your feedback and will provide a clearer motivation for our choice of baseline/reference models in the revised text. Our motivations are also briefly listed here:
>
> **Task 1:** Supervised Fine-grained NER: For the supervised NER task we used DeBERTa-v3 which is a modern common usage transformer used by millions of users (based on HF monthly downloads of model variants). In our prior experience, the DeBERTa-v3 model outperforms other BERT variants, such as RoBERTA and BERT on entity-related tasks. Its superiority to other BERT variants, both in NER and in other NLP tasks, is also documented in [1,2].
>
>
> **Task 2:** Zero Shot Fine-Grained NER with LLM: to the best of our knowledge, the best performing approach for zero-shot NER today is using an instruct-tuned LLM. At the time of writing, and arguably also today, GPT3.5 and GPT4 were among the strongest available instruct-tuned LLMs, thus we believe they represent the strongest available models for this task.  Our experiments show that zero-shot-NER is a challenging task also for LLM, and which is still far from being solved. This ties to the theme of our paper: let's take this research field to the next step.
>
> **Task 3:** Exhaustive Typed-Entity Mention Retrieval - In a zero-shot configuration, we evaluated two state-of-the-art IR vector embedding models, e5-v2 and gtr, in both their base and large variants. The outcomes are detailed in Table 2. We chose these models as they are the best-performing models in the entity-retrieval task for Dbpedia-entity-v2 and were also among the top-performing models in the IR tasks benchmark of MTEB[3].
>
> We will explicitly specify in the paper the baseline models for each task in the dataset, grounded on the results we have presented.
>
> [1] He, Pengcheng, Jianfeng Gao, and Weizhu Chen. "Debertav3: Improving deberta using electra-style pre-training with gradient-disentangled embedding sharing." arXiv preprint arXiv:2111.09543 (2021).
>
> [2] https://ibm.github.io/model-recycling/Rankings.html
>
> [3] https://huggingface.co/spaces/mteb/leaderboard
>
>
> **Clarification questions in the presentation improvement section:**
>
> Thank you for pointing out these topics, we will clarify them better in the text. Here are detailed explanations.
>
> **Q1.** *300-301: "without scanning all the documents": what does this mean? How can the result be "exhaustive" if there is a priori a filter that excludes some documents to be considered?*
>
> The phrase "without scanning all the documents" doesn't imply any filtering of the content. Instead, it contrasts with the zero-shot NER approach, where each individual sample requires an inference run. In IR tasks, the standard procedure is pre-indexing the entire corpus. This allows for a quick match between a query and a paragraph based on a certain metric, such as cosine similarity, using approximate nearest neighbor search, eliminating the need to repeatedly "scan the documents" for every single query. We'll ensure clearer wording in our revised text.
>
> **Q2.** *589: why did you pick DeBERTa, and not some other BERT variant? Please add a motivation
> Please see the answers for P2, about the baseline/reference models.*
>
> **Q3.**  *603: "s.d. 0.21": what does s.d. mean? standard deviation? on what quantity is it the standard deviation exactly?*
>
> Yes, "s.d." stands for standard deviation. In our experiment to assess robustness in the supervised context, we trained a NER model for each of 20 randomly selected entity types. These models were then applied to 10,000 random dataset paragraphs, excluding those containing the target entity type, to measure the false positive (FP) rate. The average FP rate was found to be 0.25, meaning, on average, 1 in 4 paragraphs had a false positive entity. The mentioned s.d. represents the standard deviation of the FP rate across these 20 models: in some models, the FP rate was higher than 0.25, and in others it was lower.
>
>
> **Q4.** *658 etc.: "attempting to achieve an approximately uniform distribution of entity types": please clarify how your approach would indeed amount to a uniform selection over all possible types…*
>
> Since the number of mentions of each entity type is not equal, random sampling will end with a non-balanced distribution of the classes, and each paragraph can include several entity types.
> Therefore we sampled the paragraphs in a stratified manner such that all entity types will be represented with an equal number of items. We will clarify it in the paper.
>
> We truly appreciate the time and effort you dedicated to improving the quality of the presentation.

---

### Official Review · Reviewer_jvxe · 2023-08-04

**Soundness:** 3

**Excitement:**

3: Ambivalent: It has merits (e.g., it reports state-of-the-art results, the idea is nice), but there are key weaknesses (e.g., it describes incremental work), and it can significantly benefit from another round of revision. However, I won't object to accepting it if my co-reviewers champion it.

**Paper Topic And Main Contributions:**

The first part of this paper discusses the future direction of named entity recognition (NER) in response to the emergence of large-scale language models (LLMs). Specifically, the authors list fine-grained supervised NER and zero-shot fine-grained NER, exhaustive typed-Entity retrieval, as tasks to be addressed in the future. In the latter half of the paper, NERetrieve, a dataset consisting of around 4 million paragraphs from English Wikipedia, is constructed as data that can be used to evaluate the tasks mentioned in the first part of the paper, and the results of some simplified experiments using this data are reported.

**Questions For The Authors:**

- I think it is good that it is a position paper and that it is mainly subjective, but in the current situation where many LLMs have a multilingual nature, is it appropriate to go in the direction of data construction and validation targeting only English?

**Reasons To Accept:**

- It systematically summarizes NER-related tasks after the emergence of large-scale language models (LLMs).
- Based on a large silver-annotated corpus constructed by the authors, the accuracy of several NE-related tasks has been reported.

**Reasons To Reject:**

- Even considering that it is a position paper, it is unclear what the main contribution of the paper is. It would seem that the main contribution is a discussion of NER-related tasks that will become mainstream in the future, but similar efforts have already been made for two of the proposed tasks, fine-grained supervised NER and zero-shot fine-grained NER, and they do not seem to be particularly new.
- As for the construction of the silver-annotated corpus of 4 million paragraphs covering 494 entity types, it is not clear whether this is a temporary corpus for preliminary research or a corpus that will be made publicly available in the future. In the latter case, the quality evaluation is only 91% for precision, which does not provide sufficient information to determine the actual quality of the corpus, and its usefulness as a corpus cannot be confirmed.

**Reproducibility:**

2: Would be hard pressed to reproduce the results. The contribution depends on data that are simply not available outside the author's institution or consortium; not enough details are provided.

**Reviewer Confidence:**

3: Pretty sure, but there's a chance I missed something. Although I have a good feel for this area in general, I did not carefully check the paper's details, e.g., the math, experimental design, or novelty.

**Typos Grammar Style And Presentation Improvements:**

- Line 169: in Section 7 The next -> in Section 7. The next-
- Line 506: graph knowledge base,( -> graph knowledge base (

---

> ### Author Rebuttal · Authors · 2023-08-29
>
> **Thank you for acknowledging our systematic summary of NER-related tasks post-LLM emergence.**
>
> Please see our response below:
>
> **Regarding your comment:** *“As for the construction of the silver-annotated corpus of 4 million paragraphs covering 494 entity types, It is not clear whether this is a temporary corpus for preliminary research or a corpus that will be made publicly available in the future”:*
>
> Apologies for any oversight on our part, but as we wrote in the manuscript, our dataset is derived from open-source resources (Wikipedia, DBpedia, Wikidata, Calligraph), and is intended for public use. We plan to release it, including task-specific splits, post-anonymity period. All experiments and dataset construction are reproducible, and we'll share the dataset, code, and details on our project's GitHub.
>
> **Regarding your comment:** *“The quality evaluation is only 91% for precision, which does not provide sufficient information to determine the actual quality of the corpus, and its usefulness as a corpus cannot be confirmed.”*
>
> First, since the time of our initial submission, we further worked on the dataset quality, improving its accuracy from 91% to 94%. This enhancement was achieved through a series of incremental adjustments such as improvements in the corpus parsing process, incorporating another field of aliases from Wikidata, and minor changes in the data sampling heuristics, effectively reducing potential noise generators in the dataset.
>
> We do not consider this dataset to be a preliminary resource, but a dataset created with comprehensive efforts with the highest standards which is expected for a new NLP resource.  We found your concern to be important and we will clarify and expand the discussion around the dataset quality assessment.
>
> Our manual evaluations now indicate an accuracy of 94% for the NER labels on 250 random paragraphs, a marked improvement from the earlier 91%. It's worth highlighting that this accuracy is either on par with or surpasses established manually annotated datasets. For instance, **TACRED** reports an annotation accuracy of 93.3% [1], and **SQuAD** registers an 86.8% F1 score in human annotator validation [2].
>
> To put this number in context, it is useful to compare to other large-scale NER datasets:
>
> (a) **FewNERD**, perhaps the largest and highest quality human annotated NER resource, claims a verified accuracy of 95%. Compared to FewNERD, our dataset has about 7x the number of classes, and more than 20x the number of sentences[3]. Given this, we believe our estimated accuracy level of 94% compares favorably.
>
> (b) **MultiCoNER**, a large-scale silver annotated multilingual NER dataset with 6 coarse entity types and around 217K original sentences (we have about x84 more entity types and more than 20x original paragraphs). They report manual quality assessment of the NER labels for 94% accuracy [4]. Again, given our substantially larger and more challenging entity types, we consider our accuracy to compare favorably.
> Both of these datasets are well-established works in the modern NER community.
>
> To further gauge the utility of the dataset, we reported several experiments on different NE-tasks (Supervised setup, zero shot setup,  NE mention retrieval) using our NERetrieve dataset, and for each task, our results are aligned with the trends observed in other research lines which experiment with much more simplified version of NE tasks. (see related works regarding each task). This shows the dataset indeed has discriminatory power.
>
> **Regarding your comment:** *“Even considering that it is a position paper, it is unclear what the main contribution of the paper is. It would seem that the main contribution is a discussion of NER-related tasks that will become mainstream in the future, but similar efforts have already been made for two of the proposed tasks, fine-grained supervised NER and zero-shot fine-grained NER, and they do not seem to be particularly new.”*
>
> First, we note that we see our contribution as consisting of two parts: (a) the position: discussing the role of NER in the era of LLMs, showing several directions in which NER tasks could evolve and increase in complexity, and **demonstrating empirically** that even state-of-the-art LLMs are far from solving these harder NER tasks, while still giving hope that we can progress in this direction. (b) providing a concrete dataset that supports research on the NER directions we propose.
>
>
> In our paper, we discuss various phenomena that, to our knowledge, have been rarely addressed before. We illustrate these phenomena using our unique dataset. For instance, we examine the mix of intersectional granularities in the same task—such as identifying entity-type "Writer" from different nationalities and differentiating between professions. We also evaluate the performance of supervised models as they tackle increasing granularity in entity types, illustrated by the example experiment of  ANIMAL → INVERTEBRATE → INSECT → BUTTERFLY in Figure 1. Furthermore, we assess the capabilities of foundation models in extracting fine-grained entities across diverse domains in a zero-shot setup, thereby addressing real challenges that we foresee in the future of NER. We were able to conduct these experiments due to the comprehensive nature of the dataset we have constructed. Additionally, we introduce the task of exhaustive retrieval of entity mentions, which we believe represents a significant advancement in NER tasks. This task is not only novel but also unsupported by existing datasets.
>
>
> **Regarding your question:** *“I think it is good that it is a position paper and that it is mainly subjective, but in the current situation where many LLMs have a multilingual nature, is it appropriate to go in the direction of data construction and validation targeting only English?”*
>
> As researchers native to a low-resource language, we deeply recognize the importance of multilingual NLP technologies. Indeed, we believe that our “position” part of the paper holds also for non-Englsh: the future of NER is relevant for all languages. However, given the effort and resources required to create the English NERetrieve dataset we present in this work, we find it infeasible to produce both an English and a multilingual dataset in the same work, while having them both be of high quality. We thus prioritized English for this initial effort (as is common in many other works), and reserve the multilingual version for future work, by us or others.
>
> We recognize that this response is extensive, but we believe it's crucial to clarify key points. Thank you for taking the time to consider our arguments.
>
> [1] Zhang, Yuhao, et al. "Position-aware attention and supervised data improve slot filling." Conference on Empirical Methods in Natural Language Processing. 2017.
>
> [2] Rajpurkar, Pranav, et al. "Squad: 100,000+ questions for machine comprehension of text." arXiv preprint arXiv:1606.05250 (2016).
>
> [3] Ding, Ning, et al. "Few-nerd: A few-shot named entity recognition dataset." arXiv preprint arXiv:2105.07464 (2021).
>
> [4] Malmasi, Shervin, et al. "Multiconer: a large-scale multilingual dataset for complex named entity recognition." arXiv preprint arXiv:2208.14536 (2022).

---

### Official Review · Reviewer_Yuir · 2023-08-05

**Soundness:** 3

**Excitement:**

3: Ambivalent: It has merits (e.g., it reports state-of-the-art results, the idea is nice), but there are key weaknesses (e.g., it describes incremental work), and it can significantly benefit from another round of revision. However, I won't object to accepting it if my co-reviewers champion it.

**Paper Topic And Main Contributions:**

This paper discusses the NER task in the LLMs era and talks about the next possible directions of next germination of NER tasks. They define four versions of NER task formulation in LLMs era and present a silver-annotated dataset.

**Reasons To Accept:**

The paper proposed four challenges for NER tasks in the LLM era which are important to the future research. Theses points of view are interesting to the research community and provide valuable intuitions for future research.

**Reasons To Reject:**

There are some typos in the texts, it would be better the authors could improve the writing quality of this paper. Considering this as a position paper, I am not quite sure what's is the main contribution of this paper. The paper presents a silver-annotated dataset, but does not provide a baseline model on the dataset. It would be interesting to see the performance of a baseline model on the presented dataset and lead the improvement for further investigations.

**Reproducibility:**

N/A: Doesn't apply, since the paper does not include empirical results.

**Reviewer Confidence:**

4: Quite sure. I tried to check the important points carefully. It's unlikely, though conceivable, that I missed something that should affect my ratings.

---

> ### Author Rebuttal · Authors · 2023-08-29
>
> **Thank you for finding our ideas interesting to the research community and providing valuable intuitions for future research!**
>
> **Regarding your comment:**  *Considering this as a position paper, I am not quite sure what's is the main contribution of this paper:*
>
> While the nature of a position paper is often to spotlight key areas of interest for the research community, our paper aims to both highlight pertinent issues and provide empirical evidence to support our assertions. So this could be its own contribution. However, we believe our paper also goes beyond that. Let us try and explain what we view as our contributions in this work:
> First, in terms of the position expressed in the paper, we ask what is the role of named-entity-recognition tasks in the era of LLMs. We hear many voices that consider NER tasks being solved by LLMs, and we argue this is not the case: while the “standard” NER settings might be solved, we argue that the standard tasks were simple because this is what we could solve at the time, and that we now can take NER to the next level. We suggest several concrete directions as to what such next levels may be. Empirically, we present a series of experiments showing that current “standard” models, as well as LLMs, are far from solving these more challenging NER tasks. We hope this convinces readers that there is a lot to be researched around NER, also in this “new era” of LLMs.
>
> Second, beyond expressing this position and introducing the NER tasks, we also provide a concrete dataset to facilitate research on these tasks. The dataset is designed to support the tasks in the paper including specific train/test splits for each. This is a large-scale, fine-grained NER dataset featuring a variety of domains, 494 entity types including intersectional entity types, long-tail entity types, and hierarchical relations between some (but not all) entity types. As far as we know, our dataset features the largest number of entity types available in a fine-grained NER dataset. Despite its silver-annotation procedure, we establish it to be of high quality. We hope this dataset will facilitate and promote research on future of NER.
>
> While this is a position paper, it is one grounded in empirical experiments, and which presents a dataset with the potential to significantly advance the field of NER during the LLM era.
>
> **Regarding your comment:** *The paper presents a silver-annotated dataset, but does not provide a baseline model on the dataset:*
>
> Part of our claim is that there are currently no good models for this set of future-looking tasks. Indeed, for the few-shot and zero-shot tasks, the best models are currently OpenAI’s LLMs, which are infeasible to run at the scale required by the size of the dataset. Thus, instead of providing comprehensive results for these tasks, we opted to demonstrate GPT3.5 and GPT4’s performance on a subset of 1000 paragraphs, showing that these LLMs are far from solving the tasks. For the supervised case we chose to highlight failure cases which we thought to be of interest (e.g., degrading performance with increasing type granularity), but we can also use the same DeBERTa-based models as a baseline for the supervised task, and report supervised baseline results on all classes for camera ready. Finally, for the retrieval task, we evaluated two state-of-the-art IR vector embedding models, e5-v2 and GTR, in both their base and large variants. Their results are detailed in Table 2. We consider them as a baseline for this task. We will make the baseline discussion clearer in camera ready.
>
> Lastly, we'd like to mention that we have revised the manuscript, corrected the typos, and improved the overall writing quality.

---

### Meta-Review · Area_Chair_DXcb · 2023-09-19

**Recommendation:** 3

**Metareview:**

This position paper presents a forward-looking study of LLMs over next-level NER research by introducing three new challenges including fine-grained, zero-shot based on labels, and retrieval for all instances for the query entity NER tasks. Reviewers appreciate that it presents insightful directions for future LLM and NER research. Reviewers share the concern and confusion about the paper contribution: while it is a position paper, it also provides a silver-annotated dataset but no baseline; the quality and usefulness of the silver-annotated dataset are uncertain; two out of three tasks are already explored in previous work. The authors provide effective responses to address these concerns. Overall, we strongly encourage the authors to incorporate the feedback to improve the paper.

---

### Decision · Program_Chairs · 2023-10-07

**Decision:**

Accept-Findings

**Comment:**

This position paper presents a forward-looking study of LLMs over next-level NER research by introducing three new challenges including fine-grained, zero-shot based on labels, and retrieval for all instances for the query entity NER tasks. Reviewers appreciate that it presents insightful directions for future LLM and NER research. Reviewers share the concern and confusion about the paper contribution: while it is a position paper, it also provides a silver-annotated dataset but no baseline; the quality and usefulness of the silver-annotated dataset are uncertain; two out of three tasks are already explored in previous work. The authors provide effective responses to address these concerns. Overall, we strongly encourage the authors to incorporate the feedback to improve the paper.